

# Three-dimensional visualisation of the internal anatomy of the sparrowhawk (*Accipiter nisus*) forelimb using contrast-enhanced micro-computed tomography

Fernanda Bribiesca-Contreras and William I. Sellers

Faculty of Science and Engineering, University of Manchester, Manchester, UK

## ABSTRACT

**Background:** Gross dissection is a widespread method for studying animal anatomy, despite being highly destructive and time-consuming. X-ray computed tomography (CT) has been shown to be a non-destructive alternative for studying anatomical structures. However, in the past it has been limited to only being able to visualise mineralised tissues. In recent years, morphologists have started to use traditional X-ray contrast agents to allow the visualisation of soft tissue elements in the CT context. The aim of this project is to assess the ability of contrast-enhanced micro-CT ($\mu$CT) to construct a three-dimensional (3D) model of the musculoskeletal system of the bird wing and to quantify muscle geometry and any systematic changes due to shrinkage. We expect that this reconstruction can be used as an anatomical guide to the sparrowhawk wing musculature and form the basis of further biomechanical analysis of flight.

**Methods:** A 3% iodine-buffered formalin solution with a 25-day staining period was used to visualise the wing myology of the sparrowhawk (*Accipiter nisus*). $\mu$CT scans of the wing were taken over the staining period until full penetration of the forelimb musculature by iodine was reached. A 3D model was reconstructed by manually segmenting out the individual elements of the avian wing using 3D visualisation software.

**Results:** Different patterns of contrast were observed over the duration of the staining treatment with the best results occurring after 25 days of staining. Staining made it possible to visualise and identify different elements of the soft tissue of the wing. Finally, a 3D reconstruction of the musculoskeletal system of the sparrowhawk wing is presented and numerical data of muscle geometry is compared to values obtained by dissection.

**Discussion:** Contrast-enhanced $\mu$CT allows the visualisation and identification of the wing myology of birds, including the smaller muscles in the hand, and provides a non-destructive way for quantifying muscle volume with an accuracy of 96.2%. By combining contrast-enhanced $\mu$CT with 3D visualisation techniques, it is possible to study the individual muscles of the forelimb in their original position and 3D design, which can be the basis of further biomechanical analysis. Because the stain can be washed out post analysis, this technique provides a means of obtaining quantitative muscle data from museum specimens non-destructively.

Corresponding author
Fernanda Bribiesca-Contreras,
fernanda.bribiesca@postgrad.
manchester.ac.uk

## INTRODUCTION

Gross dissection is undoubtedly the most commonly used technique for studying animal anatomy, enabling visualisation of both hard and soft tissues and the measurement of morphological features. However, dissection is a time-consuming and destructive technique that does not allow the repetition of observations once the specimen has been dissected, which often makes it unsuitable for museum specimens.

X-ray computed tomography (CT) scanning is a non-destructive alternative that has become more widespread in medical and biological studies (*Kalender, 2011*; *Mizutani & Suzuki, 2012*). Nevertheless, X-ray CT scanning of untreated specimens only permits visualisation of mineralised structures due to the low X-ray absorption capacity of soft tissue (*Mizutani & Suzuki, 2012*). Morphologists have developed new approaches for imaging soft tissue by X-ray using contrast-enhancing agents (*Gignac et al., 2016*). Contrast agents have been historically employed for studying soft tissue with light microscopy (e.g. gold and silver; *Grizzle, 1996*; *Valverde, 1970*), electron microscopy (e.g. osmium; *Mizutani & Suzuki, 2012*), and gross dissection (e.g. iodine; *Bock & Shear, 1972*). Moreover, they have been employed in diagnostic medical X-rays since the early 1900s (*Patton, 1994*) and, recently, their use has extended to X-ray CT scanning technology. A variety of agents (e.g. phosphotungstic acid (PTA), phosphomolybdenic acid (PMA), osmium tetroxide, and iodine-based solutions) have been tested in both vertebrates and invertebrates and have proved to be a suitable tool for soft tissue visualisation of fishes (*Metscher, 2009a*; *Descamps et al., 2014*), amphibians (*Metscher, 2009a*; *Kleinteich & Gorb, 2015*), reptiles (*Tsai & Holliday, 2011*; *Gröning et al., 2013*), birds (*Düring et al., 2013*; *Lautenschlager, Bright & Rayfield, 2014*; *Li et al., 2015*; *Hieronymus, 2016*), mammals (*Cox & Jeffery, 2011*; *Jeffery et al., 2011*; *Stephenson et al., 2012*; *Aslanidi et al., 2013*; *Baverstock, Jeffery & Cobb, 2013*; *Cox & Faulkes, 2014*; *Shearer et al., 2014*), vertebrate embryos (*Metscher, 2009b*; *Degenhardt et al., 2010*; *Tahara & Larsson, 2013*; *Descamps et al., 2014*; *Gignac & Kley, 2014*), and invertebrates (*Metscher, 2009b*; *Faulwetter et al., 2013*; *Boyde et al., 2014*).

In the last decade, iodine-based enhanced contrast micro-CT ($\mu$CT) has been preferred for imaging biological organisms and has proved successful for soft tissue visualisation in multiple anatomical studies (for a review, see *Gignac et al., 2016*). However, there are only a few anatomical studies of birds using iodine-based enhanced contrast $\mu$CT and none focused on the forelimb (*Düring et al., 2013*; *Lautenschlager, Bright & Rayfield, 2014*; *Li et al., 2015*), except for the work of *Hieronymus (2016)* who studied the flight feather attachments of the rock pigeon (*Columba livia*). This technique involves soaking the specimen in an iodine-based solution for a specific amount of time. The most common iodine-based solution is Lugol's iodine, or iodine-potassium iodide ($I_2KI$; *Gignac et al., 2016*). It became increasingly used after being tested on embryos, vertebrates, and invertebrates in a comparative study of the effectiveness of multiple staining agents for enhancing contrast of soft tissues (*Metscher, 2009b*). Other staining agents have achieved

similar results, or even slightly better (e.g. PTA; *Metscher, 2009a*; *Pauwels et al., 2013*; osmium tetroxide; *Metscher, 2009a, 2009b*; *Mizutani & Suzuki, 2012*), however, iodine has the advantages of being easily available, relatively inexpensive, easy to manipulate, and shows strong affinity for soft tissue (*Metscher, 2009b*; *Gignac et al., 2016*). In particular, iodine-based contrast-enhanced µCT has been commonly employed for investigating skeletal muscles as it has been observed that iodine has a strong affinity for muscle fibres and muscle fascia (*Cox & Jeffery, 2011*; *Tsai & Holliday, 2011*; *Aslanidi et al., 2013*; *Baverstock, Jeffery & Cobb, 2013*), making this technique a suitable approach for three-dimensional (3D) reconstruction of musculature that can be useful for further biomechanical analysis (*Baverstock, Jeffery & Cobb, 2013*; *Gröning et al., 2013*; *Kleinteich & Gorb, 2015*). Likewise, it facilitates the measurement of muscle parameters such as volume (*Baverstock, Jeffery & Cobb, 2013*) and fascicle length (*Jeffery et al., 2011*) that can otherwise be difficult to estimate accurately.

For iodine-based CT scanning, different methodologies have been followed depending on the sample type (e.g. taxa, developmental stage, specimen size, tissue type), specimen treatment prior to staining, staining duration, and contrast agent properties. Some authors have chosen different solutions for dissolving $I_2KI$ such as ethanol ($I_2E$; e.g. *Stephenson et al., 2012*; *Silva et al., 2015*), methanol ($I_2M$; *Metscher, 2009b*), or water ($I_2KI$ or Lugol's solution; for review, see *Gignac et al., 2016*). However, it has been demonstrated that ethanol increases tissue shrinkage, thus is not recommended when morphological measurements are fundamental (*Vickerton, Jarvis & Jeffery, 2013*). Lower or higher concentrations of iodine have been preferred and it seems that the size of the specimen to be imaged is relevant when deciding the solution concentration and duration of the treatment. As the specimen gets smaller, the distance that the iodine has to diffuse in order to reach deeper structures decreases; therefore, a lower concentration of $I_2KI$ is required and a shorter staining duration can be used (*Gignac et al., 2016*). The opposite occurs in larger specimens, which demand higher $I_2KI$ concentration and longer staining durations. However, decomposition of tissue can occur after long staining periods in larger specimens. *Li et al. (2015)* employed a protocol for imaging larger specimens over a longer duration using a modified iodine-buffered formalin solution at a low concentration. This avoids tissue damage despite the longer staining period and the lower concentration helps to minimise shrinkage (*Degenhardt et al., 2010*; *Vickerton, Jarvis & Jeffery, 2013*).

The aim of the present work is to test the utility of iodine-based contrast-enhanced µCT for studying the myology of the avian forelimb, since the diffusion distance is relatively large and the appropriate protocol for wing soft tissue is currently unknown, and to build a 3D model of the internal anatomy of the avian wing. The process was undertaken using a sparrowhawk (*Accipiter nisus*) wing as a model, which was chosen as a representative of birds of prey and birds of a similar size. A low concentration iodine-buffered formalin solution was preferred to avoid tissue damage, as the necessary staining duration was uncertain; CT scans were taken at stepped time increments during staining. In addition, numerical data of muscle geometry was obtained for comparison with quantitative dissections to assess the impact of shrinkage on muscle volumes.

We expect that the findings of this work will be useful for further studies of the functional morphology of wing musculature and its implication in flight biomechanics. Furthermore, we expect that this project will contribute to our knowledge of the forelimb myology of Accipitriformes, which is only available for the shoulder and arm musculature of the common buzzard (*Buteo buteo*), European honey buzzard (*Pernis apivorus*) (*Canova et al., 2015*) and for some wing muscles of different species in functional morphology analyses (e.g. *Nair, 1954*; *Corvidae, Bierregaard & Peters, 2006*), and that the model can be used as an anatomical and dissection guide for the sparrowhawk forelimb musculature that is currently lacking.

## MATERIALS AND METHODS

### Specimen treatment

The left wing of an adult female sparrowhawk (*A. nisus*), weighing 281.6 g with a wing length of 19.2 cm (distance from the wrist joint to the tip of the folded wing; *Pyle, 1997*), was used for this study. The specimen was donated from the World Museum of Liverpool and kept frozen at −20 °C before use, then left to thaw overnight before dissection. The wing was dislocated free from the shoulder by cutting the tendons and fleshy attachments of the shoulder and pectoral muscles. After being plucked and skinned, it was fixed in 10% neutral-buffered formalin (NBF) and stored in a fridge at 3 °C for two weeks. For staining, it was transferred to a ~3% (w/v) iodine-buffered formalin solution in an enclosed jar. The staining solution was prepared using the method described by *Li et al. (2015)* for larger specimens (8 g iodine, 16 g potassium iodide into 800 ml 10% NBF). The choice of a lower concentration was made to avoid tissue shrinkage, as it has been previously observed that soft tissue shrinkage increases with higher concentrations of iodine (*Tahara & Larsson, 2013*; *Vickerton, Jarvis & Jeffery, 2013*; *Gignac & Kley, 2014*).

### Imaging

For imaging, the wing was removed from the fixing and staining solution and scanned in air with a Nikon XTEK XTH 225 kV μCT system at the Henry Moseley X-ray Imaging Facility, University of Manchester. Before each scan, the wing was removed from the solution and rinsed with water to remove excess stain and prevent surface saturation. A scan of the unstained wing (70 kV, 200 μA) was taken prior to staining to act as a control image, followed by scans of the stained wing after three, 10, 15, 18, and 25 days in the staining solution (80 kV, 100 μA, 0.25 mm aluminium filter) to allow comparison of the contrast achieved for different staining durations. Each time, three separate scans were taken of the regions of interest (brachial arm, antebrachial arm, and hand) at the same position to facilitate further merging of the whole wing in a single file. A total of 5,013 projections for each scan were obtained over 360°, resulting in an acquisition time of 1 h 24 min (4 h 12 min in total for the whole wing) at a resolution of 25 μm/voxel for each region of interest.

### Reconstruction of 3D model

Reconstruction of the wing from the CT images was carried out with the Nikon Metrology NV CT-Pro reconstruction software (version V4.0.5360.28810, September 2014) and imported into Avizo (version Lite 9.0; Visualisation Centre Group) for further visualisation, merging of the regions of interest and segmentation of individual anatomical elements. To merge the brachial arm, antebrachial arm, and hand into a single file, the volume rendering of the three sections were aligned and merged together by applying the 'Merge' module. The scans of the stained wing after 25 days were used for identification and segmentation of individual muscles. Each muscle and bone was manually segmented out using both the 'magic wand' and 'painting brush' tools, and interpolating between five and 10 slices. This was preferred over the 'threshold' option due to the similarity in grey-scale intensity between bones and muscles as a consequence of the radiodensity reached during the staining treatment, although they could be clearly distinguished from each other by a human operator, as reported in similar studies (*Cox & Jeffery, 2011*; *Baverstock, Jeffery & Cobb, 2013*; *Cox & Faulkes, 2014*; *Lautenschlager, Bright & Rayfield, 2014*). Isosurfaces of the elements segmented out were generated to build a 3D model of the wing in order to visualise and identify the hard and soft tissue. Furthermore, isosurfacing allows the calculation of the volume of each muscle by means of the Avizo 'measure and analyse module'. Muscle volumes calculated from CT data were compared to volumes obtained from gross dissection (calculated from the muscle mass and a standard value for vertebrate muscle density [1.06 g·cm$^{-3}$; *Mendez & Keys, 1960*; *Brown et al., 2003*]) from both the scanned wing (after the stain was washed out) and the fresh right wing of the same individual by linear regression analysis using R v. 3.3.2 (https://www.r-project.org). This comparative analysis was performed for assessing the accuracy of the wing model to obtain quantitative data of muscle architecture (scanned wing vs. 3D model) and to quantify any shrinkage due to staining (fresh wing vs. 3D model). To remove the stain after the last scan was taken, the stained wing was transferred to 10% NBF in an enclosed jar and stored in a fridge at 3 °C. After 48 days, the wing returned to its original colour, then it was washed with water and stored in the fridge overnight prior to dissection. The nomenclature of skeletal structures and musculature of the sparrowhawk wing followed in this study is consistent with *Nomina Anatomica Avium* (*Baumel et al., 1979*)

## RESULTS

Different patterns of contrast were observed during the 25 days of the staining procedure, as they are shown in Fig. 1. The control image (Fig. 1A) only showed detail of the bones while the distinct elements of the soft tissue are unrecognised. An improvement of contrast is evident as the duration of staining increases (Figs. 1B–1F). Moreover, it can be observed that it takes longer for the iodine solution to reach the deeper muscles of the brachial region (Fig. 1; first row), where clear visualisation of all the muscles is not achieved until the last day of the treatment. In comparison, the smaller muscles of the distal wing (Fig. 1; third row) were stained after only three days of treatment.

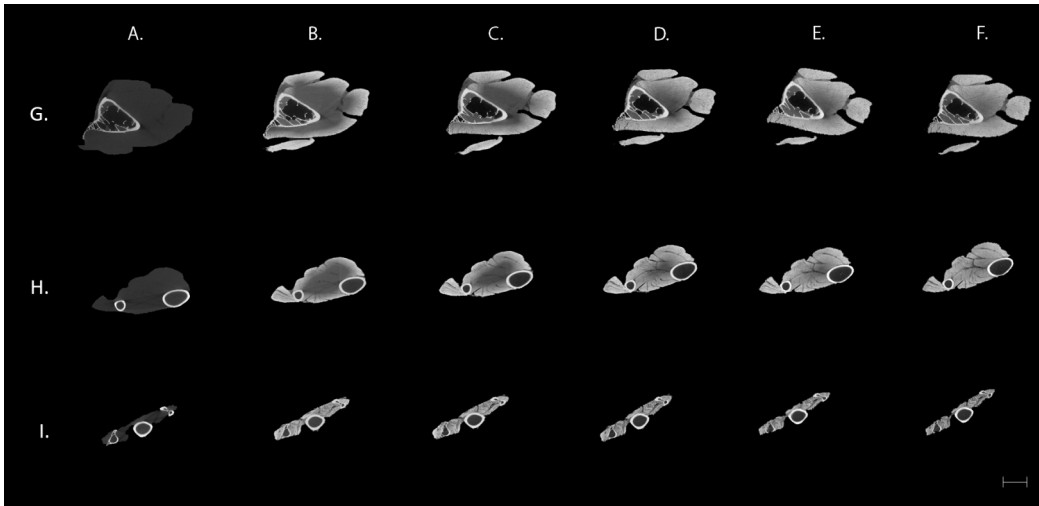

**Figure 1 Transverse µCT images of a sparrowhawk (*A. nisus*) wing.** Columns: (A) Control. (B–F) ~3% (w/v) iodine-buffered formalin solution after three (B), 10 (C), 15 (D), 18 (E), and 25 (F) days. Scale bar equal to 5 mm. Rows: (G) Corresponds to brachial area, (H) antebrachial area and (I) to the avian hand.

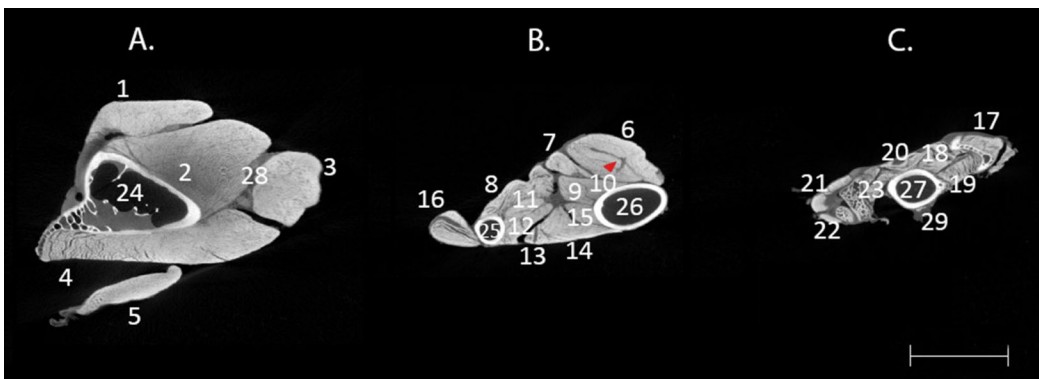

**Figure 2 Difference of contrast in transverse µCT images of *A. nisus* wing.** Transverse µCT images of *A. nisus* wing stained in a ~3% (w/v) iodine-buffered formalin solution for 25 days showing hard and soft tissue elements of the brachial region (A), antebrachial region (B), and hand (C). Red arrow shows an area of low attenuation corresponding to the internal tendon of FCU (6). Scale bar equal to 5 mm. 1, BB; 2, ST; 3, HT; 4, DMA; 5, TPLA; 6, FCU; 7, FDS; 8, PP; 9, FDP; 10, UMV; 11, ELDM; 12, ELA; 13, EDC; 14, ECU; 15, ECTU; 16, EMR; 17, UMD; 18, ISD; 19, ISV; 20, ABDM; 21, ABA; 22, EBA; 23, ADA; 24, HUM; 25, R; 26, U; 27, CMC; 28, fascia; 29, skin. Abbreviations: ABA, *abductor alulae*; ABDM, *abductor digiti majoris*; ADA, *adductor alulae*; BB, *biceps brachii*; BR, *brachialis*; CCr, *coraco-brachialis cranialis*; CMC, *carpometacarpus*; DMA, *deltoides major*; EBA, *extensor brevis alulae*; ECTU, *ectepicondylo ulnaris*; ECU, *extensor carpi ulnaris*; EDC, *extensor digitorum communis*; ELA, *extensor longus alulae*; ELDM, *extensor longus digiti majoris*; ELDMd, *extensor longus digiti majoris pars distalis*; EMR, *extensor metacarpi radialis*; FA, *flexor alulae*; FCU, *flexor carpi ulnaris*; FDMI, *flexor digiti minoris*; FDP, *flexor digitorum profundus*; FDS, *flexor digitorum superficialis*; HT, *humerotriceps*; HUM, humerus; ISD, *interosseus dorsalis*; ISV, *interosseus ventralis*; PP, *pronator profundus*; PS, *pronator superficialis*; r, radiale; R, radio; ST, *scapulotriceps*; SU, *supinator*; TPLA, *tensor propatagialis pars longa*; U, ulna; u, ulnare; UMD, *ulnometacarpalis dorsalis*; UMV, *ulnometacarpalis ventralis*.

Furthermore, the contrast-enhanced µCT images of the bird wing demonstrate different attenuation among the tissues. Figure 2 shows more detailed images of the anatomical sections of the wing where it is possible to recognise elements of the

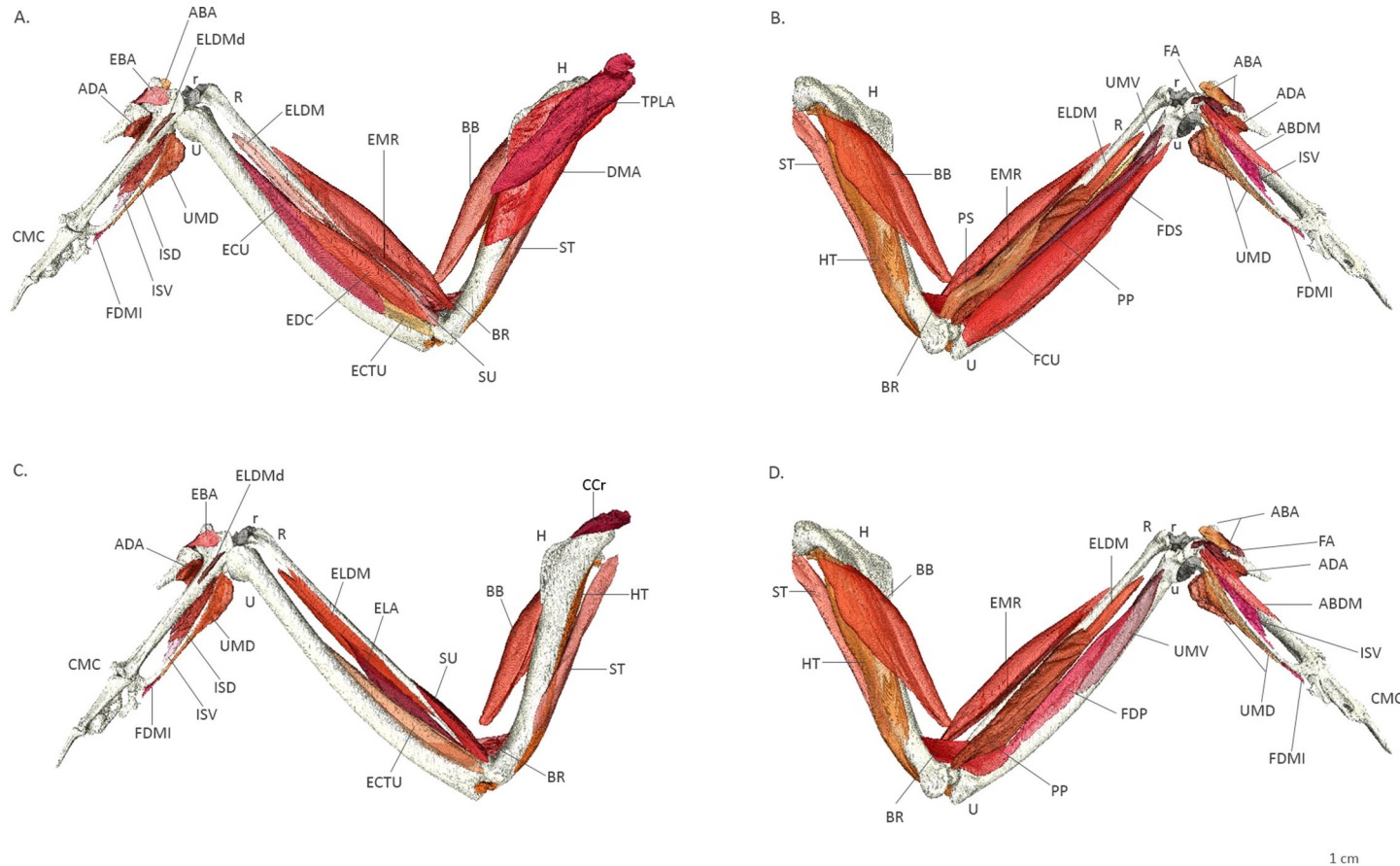

**Figure 3 Three-dimensional model of the wing muscles of a sparrowhawk.** Three-dimensional model of the wing muscles of a sparrowhawk reconstructed from CT images of the stained wing after 25 days in a ~3% iodine-buffered formalin solution. Dorsal view of superficial (A) and deep (C) muscles and ventral view of superficial (B) and deep (D) muscles. Abbreviations as indicated in Fig. 2.

connective tissue separating the muscles and remains of the skin in the hand. Iodine in this buffer seems to have a low affinity for ligaments and tendons as it can be observed in the muscle *flexor carpi ulnaris* (FCU) where the dark area corresponds to the internal tendon of this muscle (red arrow in Fig. 2B).

A 3D model of the musculoskeletal system of the sparrowhawk wing is presented in Fig. 3 and a rotating animation in File S1 showing the 30 muscles comprising the avian forelimb. However, it also illustrates that little stain is taken up by other soft tissues and in particular reinforces the observation of an almost complete lack of staining in tendons (e.g. distal attachment of *biceps brachii*).

A linear regression analysis of the muscle volumes calculated from dissection, of the fresh and scanned wings, and obtained from the 3D model showed a significant correlation ($R^2 = 0.96$, Fig. 4). The slope of the scanned wing vs. 3D model (grey line) was 0.8603, validating the use of 3D imaging techniques for obtaining quantitative data of muscle architecture, and 0.6366 for the fresh wing vs. 3D model (black line), indicating that a consistent degree of shrinkage (36.34%) for muscles of different sizes occurred after the duration of the staining. This can be clearly observed by the increase in the gaps

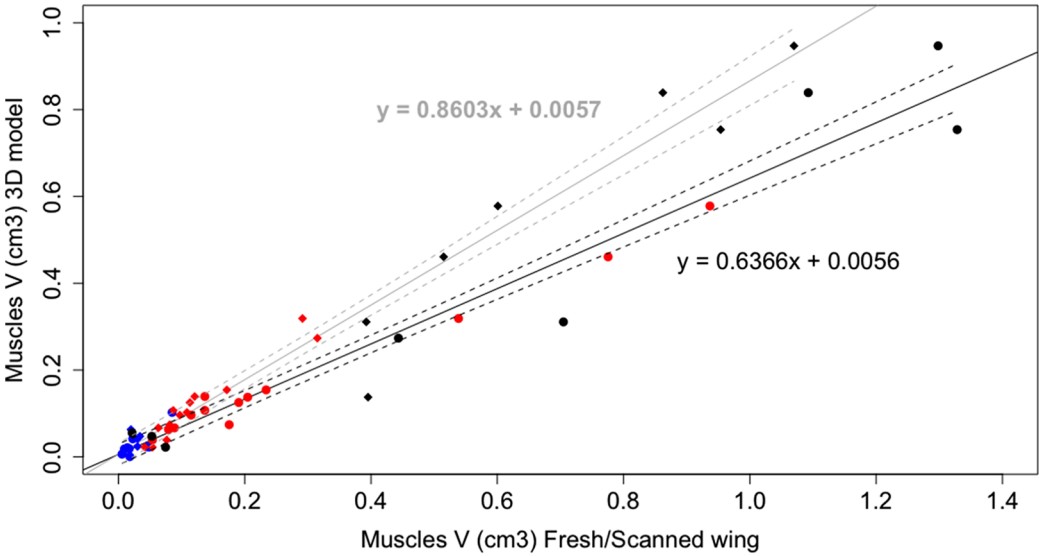

**Figure 4 Linear regression plot of wing muscle volumes.** Linear regression plot of the wing muscle volumes of the fresh wing (black lines, solid circles) and scanned wing (grey lines, solid diamonds) against the wing muscle volumes of the 3D model. Solid lines correspond to the regression lines and dashed lines to the 95% prediction intervals. Blue: hand muscles, red: antebrachial muscles, black: brachial muscles.

between adjacent muscles in Fig. 1 over the duration of the treatment. The resulting slopes were significantly different from 1 ($p < 0.001$). A table of muscle volumes for the fresh wing, scanned wing and 3D model is provided in File S2.

## DISCUSSION

Contrast-enhanced μCT using an iodine-buffered formalin solution produces highly detailed visualisation of the muscles forming the avian wing. In the case of the hand, full contrast was reached during the first three days of staining (Fig. 1B). In contrast, it took much longer to stain the larger brachial musculature. After two weeks, individual muscles can be identified although the dark area near the skeleton indicates that the iodine has not yet reached the deepest fibres of the larger muscles, such as the *scapulotriceps* (ST) (Fig. 1D). This differential improvement of contrast during the duration of the treatment illustrates the relationship between specimen size and staining, which is important to consider when designing a staining protocol (*Tahara & Larsson, 2013*; *Li et al., 2015*; *Gignac et al., 2016*).

The technique was effective for revealing the individual muscles of the sparrowhawk wing. The contrast agent showed a strong affinity for the muscle fascia and muscle fibres allowing the identification of individual muscles and visualisation of the fibre orientation in the tiny muscles of the hand (Figs. 1 and 2). It is believed that iodine adheres to different constituents of tissues, such as glycogen and lipids (*Bock & Shear, 1972*), and that it has a strong affinity for muscle fascia and muscle fibres (*Jeffery et al., 2011*; *Tsai & Holliday, 2011*; *Baverstock, Jeffery & Cobb, 2013*) and internal structures of ligaments and tendons (*Shearer et al., 2014*). However, our results do not show the
insertion and origin tendons of the wing muscles, indicating poor affinity of the iodine for these structures. Since tendon morphology is a significant component for biomechanical analysis, this is an important consideration if tendon visualization is the primary goal of the study (*Sellers et al., 2010*). It is possible to visualise the internal tendons of some muscles as dark areas of low attenuation inside the muscles such as in the FCU (Fig. 2B). Using an alternative contrast agent, such as PTA and PMA that are known to bind to collagen fibres (*Mizutani & Suzuki, 2012*; *Pauwels et al., 2013*; *Descamps et al., 2014*; *Shearer et al., 2014*), may help us to visualise tendinous structures of the wing muscles of the sparrowhawk. This was recently achieved by *Hieronymus (2016)*, who developed a comprehensive 3D model of the internal anatomy of the rock pigeon by combining μCT scanning and histological sectioning to study the anatomical structures that shape the wing during flight. Increasing the duration of the staining treatment might improve the visibility of the wing tendons as well since it was possible to observe the surface and internal structure of the anterior cruciate ligament and patellar tendon in porcine specimens after 70 days of staining with $I_2KI$ (*Shearer et al., 2014*) but this is considerably longer than the duration of the current experiment and may be related to the diffusion distance in these larger specimens (*Gignac et al., 2016*). With prolonged staining periods, there is also a risk of the muscles being over-stained which would reduce the visualisation of other important soft-tissues (*Gignac & Kley, 2014*; *Gignac et al., 2016*).

Our results show that contrast-enhanced μCT using an iodine-buffered formalin solution is a suitable technique for visualising and identifying the different muscles forming the avian wing. By combining it with 3D visualisation techniques, it is possible to study the wing muscles in their original position, which is very difficult during gross dissections where anatomical structures are commonly damaged or precise 3D relationships are difficult to discern (*Lautenschlager, Bright & Rayfield, 2014*) and deeper structures are difficult to reach (*Cox & Jeffery, 2011*; *Cox & Faulkes, 2014*). The 3D model (Fig. 3, File S1) provided in this work can be used as an anatomical and dissection guide of the wing musculature, in particular of the smaller muscles of the hands that attach directly to the bones for which only a few illustrated descriptions are available (*Vazquez, 1995*; *Zhang & Yang, 2013*; *Yang, Wang & Zhang, 2015*; *Hieronymus, 2016*); however, care must be taken due to the lack of tendons in this reconstruction and this is an area where more work is clearly required.

Furthermore, this model clearly illustrates the value of contrast-enhanced μCT for reconstructing the 3D shape of individual muscles and their anatomical relations, which is essential for biomechanical models and functional morphology analyses (*Sellers et al., 2013*). This technique proved to be a non-destructive alternative for obtaining the quantitative muscle architecture data required for advanced biomechanical techniques such as multibody-dynamics analysis (*Jeffery et al., 2011*; *Gröning et al., 2013*; Fig. 4). Nevertheless, it is essential to normalise the data against dissection data because significant shrinkage is present even when using a lower concentration of iodine (*Li et al., 2015*; *Gignac et al., 2016*), as we noticed after comparing our CT data with the fresh wing of the same individual during dissection where a shrinkage of 36.34% occurred after 25 days of staining (Fig. 4). Similar results have been reported for

skeletal muscle in mice with 2% I$_2$KI (*Vickerton, Jarvis & Jeffery, 2013*) and New Zealand rabbits with 3% I$_2$KI (*Buytaert et al., 2014*) that presented ~20% and 34–48% shrinkage, respectively, in muscle volume after the staining treatment. Iodine staining has the advantage of being reversible (*Bock & Shear, 1972*) supporting the use of iodine-based enhanced contrast μCT for non-destructively quantifying muscle data from museum specimens.

## CONCLUSION

Contrast-enhanced μCT has been demonstrated to be a suitable non-destructive alternative for gross dissection to study the wing musculature of birds. By using a low concentration of an iodine-buffered formalin solution for a 25-day staining period, it was possible to visualise and identify all the individual muscles of the sparrowhawk wing; however, staining of tendons was not achieved. Therefore, it is recommended to test the use of alternative contrast agents (e.g. PTA or PMA) for a full assessment of the anatomical elements forming the musculoskeletal system of the avian wing. Finally, we presented a 3D model of the internal anatomy of the sparrowhawk wing by combining contrast-enhanced μCT with 3D visualisation techniques where it is possible to see the muscle arrangement in their original anatomical position. In addition, it is possible to obtain quantitative data of muscle architecture from this model that, after normalising with numerical dissection data, can be useful for further biomechanical analysis and functional predictions of the role of individual muscles during flapping flight.

## ACKNOWLEDGEMENTS

We are grateful to Julia Behnsen and Charlotte Brassey for their help and suggestions during the CT scanning and 3D modelling and to Tony Parker for the donation of the specimen used in this project. We also thank Ben Parslew for discussion and helpful comments that improved this manuscript.

### Funding

This work was supported by a scholarship provided to Fernanda Bribiesca-Contreras by the Consejo Nacional de Ciencia y Tecnologia (CONACyT). The funders had no role in study design, data collection and analysis, decision to publish, or preparation of the manuscript.

### Grant Disclosures

The following grant information was disclosed by the authors:
Consejo Nacional de Ciencia y Tecnologia (CONACyT).

### Competing Interests

The authors declare that they have no competing interests.

## Author Contributions

- Fernanda Bribiesca-Contreras conceived and designed the experiments, performed the experiments, analysed the data, wrote the paper, prepared figures and/or tables, and reviewed drafts of the paper.
- William I. Sellers wrote the paper and reviewed drafts of the paper.

## Data Deposition

Figshare: https://figshare.com/s/4bd662fb390ba1a3807b

## Supplemental Information

Supplemental information for this article can be found online at http://dx.doi.org/10.7717/peerj.3039#supplemental-information.

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
