# Peer review of "Three-dimensional visualisation of the internal anatomy of the sparrowhawk (Accipiter nisus) forelimb using contrast-enhanced micro-computed tomography"

_PeerJ, doi:10.7717/peerj.3039_

## Round 0.1 · original submission · Minor Revisions

I have now received three reviews of your manuscript. I agree with all reviewers, that the study is very valuable but it requires more work. The reviewers provided several points that should be addressed in order to improve the ms. For the revised version of your work I would like you to highlight in the introduction section the aspects indicated by our first reviewer. It is very important that you quote the source of your anatomical nomenclature. You should also clarify how you washed out the stain of your sample. Finally, I find it very important that you pay attention to the suggestion of the change of the focus and the title of your manuscript.

Additional recommendations of all reviewers are mostly constructive, and attention to their suggestions will serve to improve the ms.

·

Basic reporting

The authors assess the ability of contrast-enhanced micro-CT to reconstruct a 3D model of the musculoskeletal system of the sparrowhawk (Accipiter nisus) wing, and quantify muscle geometry
and changes due to shrinkage. The contribution of this work is to develop a protocol for the study of soft tissues in the forelimb of adult birds, which so far is unknown.
I'm not a native English speaker, but the manuscript seems to be clear. The literature is well referenced, exhaustive and appropriate. The structure of the manuscript fits to the PeerJ standards. Figures are relevant, high quality, and well labelled and described (some minor points are detailed below).

Experimental design

The manuscript suits with the aims and scope of the PeerJ journal. The research is well defined and meaningful (although some relevant aspects of studying the wing myology in the sparrowhawk would be important to highlight, see below). Methods are described with sufficient detail.

Validity of the findings

The results are well presented and conclusions are well stated. These are interesting findings and deserve to be published.

Additional comments

Introduction
- The authors performed a detailed description of micro-CT method for studying animal anatomy (including advantages of the method, the use of contrast agents for the visualization of soft tissue, measurement of morphological features, and which protocol to follow depending on the sample type). However, I think it would be important to highlight in the Introduction section some relevant aspects of studying the wing myology in the sparrowhawk (although some are mentioned in the discussion), such as: 1- the available information of forelimbs muscles of Accipitriformes is scare in the literature, and this is the first work dealing with the wing muscles of Accipiter nisus; 2- works using contrast-enhanced micro-CT in birds are few and none focus on the forelimbs; and 3- this study provides a base information for other studies (e.g. finite element analysis, geometric morphometrics, among others) that will allow functional inferences in locomotion and hunting modes.
- I am not sure that the sparrowhawk can be defined as a model species. Perhaps it is more appropriate to say that this species can be considered particularly as representative of birds of prey (or birds of similar size), because the size of the wings is extremely variable within birds and, therefore, the protocol would also be variable.

Methods
- Please clarify the source of osteological and myological nomenclature. Usually for Aves is followed by Baumel et al. (1993). Handbook of Avian Anatomy.
- The reconstruction of the 3D model was performed from the CT images by scans of the stained wing on day 26 or by a combination of the scans of the different regions (brachial arm, antebrachial arm and hand) from days 4 to 26?. Please clarify it.
- In this section it should be mentioned that a regression analysis for the wing muscles volumes of the fresh wing and 3D model is done, and the software used to do it.

Results
- I think the following sentences are more appropriate to include in the Discussion section:
Lines 175-177: “Since tendon morphology can be an important component of biomechanical analysis this (the iodine low affinity for ligaments and tendons) is potentially an important consideration (Sellers et al., 2010)”. This sentence can be added to the line 211 and above.
Lines 179-181: “This clearly illustrates the value of this technique for reconstructing the 3D shape of the individual muscles and their anatomical relations which can be essential for biomechanical models (Sellers et al., 2013)”. This sentence can be included to the line 241 and above.
Lines 187-189: “This result (those from the regression analysis) validates the use of 3D imaging techniques for obtaining quantitative data of muscle architecture although the slope shows that there is appreciable shrinkage (~33%), which needs to be taken into consideration to obtain useful quantitative data”. This sentence can be added to the line 245 and above.
- Also a table with comparisons of muscle volumes of the fresh wing and 3D model may be useful for future studies and would be appreciated.

Conclusions
- Replace “By using a low concentration of an iodine-based buffered formalin solution for a two-week staining period, it was possible to visualise and identify all the individual muscles of the bird wing” with “By using a low concentration of an iodine-based buffered formalin solution for a two-week staining period, it was possible to visualise and identify all the individual muscles of the sparrowhawk wing”. It seems more appropriate due to the variable size of the wings within birds and the consequent possible adjustments in the protocol (see above).

Minor points
- Please use "e.g." in roman type or in italic type throughout the text.
- In Methods section says that the scans were taken on days 4, 11, 16, 19 and 26, but in the Results and Discussion sections and in the Figures 1 and 2 says 3, 10, 15, 18 and 25 days. Make sure to homogenize the days throughout the manuscript.
- In Introduction section, line 85, de Crespigny et al. (2008) is cited as an example where muscle volume can be measured with the iodine-based contrast-enhanced μCT technique, when in fact the article mentions the possibility of making accurate measurements of brain morphometric parameters such as volume (but not specifically of muscles).
- Replace the heading “Methods” with “Materials and Methods” conforms to PeerJ standards.
- In References section, Gignac et al. (2016) is already included in an issue: Gignac PM., et al. 2016. Diffusible iodine-based contrast-enhanced computed tomography (diceCT): an emerging tool for rapid, high-resolution, 3-D imaging of metazoan soft tissues. Journal of Anatomy 228: 889-909. DOI: 10.1111/joa.12449.
- In Figure 2, replace “Abbreviations as indicated in Fig. 4.” with “Abbreviations as indicated in Fig. 3.”
- In Figure 3, I recommend to list the abbreviations of the muscles in an alphabetical order (and not in a proximo-distal order) to find them easily.
- In Figure 4, replace “black: brachial muscles” with “grey: brachial muscles”.

Reviewer 2 ·

Basic reporting

This is a nice paper detailing the methodology used to carry out contrast-enhanced CT-scanning of an avian wing, with accompanying digital dissection and some basic muscle architecture metrics (muscle volumes). Good referencing of previous literature in this area as well.

There are some grammatical errors and awkward phrasing throughout the manuscript - please see the attached annotated manuscript for suggestions on how to correct these issues. There are some confusing sentences that need to be rephrased (ex., Lines 127 and 236).

Please try to avoid repeating information, there were a few instances of this (see annotated file).

Figures - see minor corrections in figure captions on the annotated file. The black arrow pointing to tendon in Figure 2 is impossible to see - please make this another color, such as yellow or red for contrast. Also, you must define anatomical abbreviations the first time you use them - you cannot use them in Figure 2 and refer the reader to Figure 4.

Experimental design

On two occasions (Lines 23 and 293) you state the specimen was stained over two weeks; however, optimal staining took 25 days. Please correct this.

You say that in order to calculate muscle shrinkage due to staining you compared muscle volumes from the CT data to those of: 1) the stained specimen once the stain had been washed out - which you never actually detail HOW you washed out the stain - and 2) the fresh, unstained right wing of the individual. You never actually make use of a comparison against #1 - what was the point of dissecting the stained specimen by hand? More importantly, there may be some question whether the left and right wings were the same size (individuals can exhibit right-left asymmetry). How would you address such a question?

Validity of the findings

Fine, no comment.

Annotated reviews are not available for download in order to protect the identity of reviewers who chose to remain anonymous.

·

Basic reporting

No comment.

Experimental design

The stated research question this report addresses is whether micro-CT imaging can be used to construct a model of bird wing musculature. However, the basic utility and application of contrast enhanced micro-CT imaging to animal morphology has already been abundantly demonstrated. Especially relevant here is the work by Holliday et al. on alligator jaws:

Holliday CM, Tsai HP, Skiljan RJ, George ID, Pathan S (2013) A 3D Interactive Model and Atlas of the Jaw Musculature of Alligator mississippiensis. PLoS ONE 8(6): e62806. doi:10.1371/journal.pone.0062806

Perhaps a more pertinent focus for this report is the anatomical model itself, and the specifics of how micro-CT imaging was used here for making such a model. This is really the most important result in this paper and should be presented as such. I suggest modifying the title and abstract to reflect this.

Other specific comments on the methods are noted in the pdf of the manuscript.

Validity of the findings

This report presents results from only one imaged specimen. The main product of the work is a digital model of Accipiter wing musculature. The reconstructed tomography image stack is shared, but the model should be also.

Other specific comments are noted in the manuscript pdf.

Additional comments

No comment.

---

## Round 0.2 · accepted · Accept

Thank you very much for your close attention to the suggestions of the reviewers. I think that this manuscript is ready for publication.